# Different Outcomes of Chicken Infection with UK-Origin H5N1-2020 and H5N8-2020 High-Pathogenicity Avian Influenza Viruses (Clade 2.3.4.4b)

**DOI:** 10.3390/v15091909

**Published:** 2023-09-12

**Authors:** Amanda H. Seekings, Caroline J. Warren, Saumya S. Thomas, Fabian Z. X. Lean, David Selden, Benjamin C. Mollett, Pauline M. van Diemen, Ashley C. Banyard, Marek J. Slomka

**Affiliations:** 1Department of Virology, Animal and Plant Health Agency (APHA), Addlestone, Surrey KT15 3NB, UK; 2Department of Pathology and Animal Sciences, Animal and Plant Health Agency (APHA), Addlestone, Surrey KT15 3NB, UK; 3WOAH/FAO International Reference Laboratory for Avian Influenza, Swine Influenza and Newcastle Disease, Animal and Plant Health Agency (APHA), Addlestone, Surrey KT15 3NB, UK

**Keywords:** avian influenza, HPAIV, chicken, infection, pathogenesis, transmission

## Abstract

Clade 2.3.4.4 H5Nx highly pathogenic avian influenza viruses (HPAIVs) of the “goose/Guangdong” lineage have caused a series of European epizootics since 2014. During autumn/winter 2020–2021, several H5Nx subtypes were detected in the UK, with H5N8 being the dominant subtype in wild birds and poultry. Despite the greater subtype diversity (due to viral neuraminidase gene reassortment) reported in wild birds, only H5N8 and H5N1 subtypes caused clade 2.3.4.4 UK HPAIV poultry outbreaks during this period. The direct inoculation of layer chickens showed that H5N8-2020 was more infectious than H5N1-2020, which supported the European H5N8 dominance during that season. However, the mean death time was longer for H5N8-2020 (3.42 days) than for H5N1-2020 (2.17 days). Transmission from directly infected to naive in-contact chickens was inefficient for both subtypes. Histological lesions, the tissue dissemination of viral antigen, and nucleic acid were more extensive and abundant and accumulated more rapidly for H5N1-2020 compared with H5N8-2020. Although inefficient, H5N1-2020 transmission was faster, with its greater virulence indicating that this subtype posed a major concern, as subsequently shown during H5N1 dominance of the clade 2.3.4.4 epizootic since autumn 2021. An evaluation of these *in vivo* viral characteristics is key to understanding the continuing poultry threats posed by clade 2.3.4.4 H5Nx HPAIVs.

## 1. Introduction

The “goose/Guangdong” (GsGd) lineage of highly pathogenic avian influenza viruses (HPAIVs) first emerged in 1996 in China and remains a continuing threat to global poultry production while also posing a zoonotic risk [1,2]. The GsGd lineage circulated as the H5N1 subtype for more than a decade, during which it diversified into a number of clades, which are based on the phylogeny of the haemagglutinin (HA (H5)) gene sequence and was accompanied by H5N1 endemicity in several Asian countries [3,4,5,6].

Further diversification, including the subsequent reassortment of the neuraminidase (NA) gene segment, led to the emergence of clade 2.3.4.4 H5N8 HPAIVs, the most significant GsGd clade since 2014 when widespread poultry farm outbreaks and wild waterfowl cases were initially reported in East Asia, followed by its dissemination through autumnal wild waterfowl (anseriforme) migration, leading to H5N8 cases in Europe during winter 2014–2015 [7]. There have been four additional clade 2.3.4.4 incursions into Europe, which are all predicated on a similar seasonal pattern. These viruses caused varying degrees of wild bird cases and poultry outbreaks in Europe, and they are also referred to as the H5Nx subtype as a result of further viral genetic reassortment events with other Eurasian avian influenza viruses (AIVs), which included NA gene exchange, resulting in minority H5N5 and dominant H5N6 subtypes during clade 2.3.4.4b incursions in 2016–2017 and 2017–2018, respectively [8,9,10].

The subsequent European clade 2.3.4.4 incursions during autumn 2020 were anticipated by the prior detection of H5N8 HPAIVs in Russia and Kazakhstan during the late summer [11], leading to an intense period of 513 wild bird cases (385 of which were identified in wild waterfowl) and 371 poultry outbreaks between October and early December 2020 [9]. While the 2020–2021 European H5Nx epizootic was dominated by the H5N8 subtype, clade 2.3.4.4b of the GsGd lineage has demonstrated its continuing propensity for reassortment, with a number of H5N5 and H5N1 wild bird and poultry cases also being identified by late 2020. The European H5N5 emerged as three genotypes that included different combinations of other Eurasian-origin PB2, PA, NP, and NA (N5) genes, which reassorted into the H5N8 genome. The single H5N1 genotype retained only the HA (H5) and M-gene segments of the contemporary European H5N8 virus, thereby representing an extensive reassortment event with the NA and five internal genes being of other Eurasian (i.e., non-clade 2.3.4.4) origin [9]. The European epizootic continued into early 2021, with the UK alone reporting a total of 18 H5N8 poultry outbreaks (plus two outbreaks in captive birds), two H5N1 poultry outbreaks plus 311 H5Nx wild bird cases by February 2021, 243 of which were identified in wild anseriformes [12].

Previous *in vivo* investigations of UK-origin clade 2.3.4.4 H5Nx HPAIVs showed these to be anseriforme-adapted, with efficient infection and onward transmission demonstrated to occur among ducks [13,14,15]. When contact galliforme species were introduced for cohousing with infected ducks, the efficiency of inter-species transmission varied depending on the galliforme host and the particular H5Nx virus; UK anseriforme-origin H5N8 viruses from 2014 and 2016 transmitted efficiently from ducks to infect all contact turkeys, and H5N8-2014 similarly transmitted to all the immediately cohoused chickens. However, H5N6-2017, while spreading efficiently among ducks, could only transmit and spread sporadically or unpredictably from infected ducks to contact chickens and turkeys, thereby reflecting the H5N6 2017–2018 European epidemiology, which was essentially restricted to wild birds (mainly anseriformes), with very few outdoor or backyard poultry cases and no outbreaks in commercial chickens or turkeys [10,14].

In this study, we investigated the ability of the clade 2.3.4.4b H5N8 and H5N1 HPAIV isolates from late 2020 to infect and transmit among chickens, the two subtypes being selected on the basis of their distinct genotypes. Previously, we demonstrated that infected ducks are the source of H5Nx environmental contamination, which may be important in enabling the spread of infection to naive introduced birds [14]. Therefore, by focusing on a purely chicken infection study, we determined the extent of viral contamination in the environment which originated from this species. In addition, the earlier clade 2.3.4.4 H5Nx inter-species transmissions from ducks to chickens and turkeys were accompanied by the emergence of amino acid polymorphisms [13,14]. A similar investigation of genetic changes was carried out with the H5N8 and H5N1 subtypes from the European 2020–2021 epizootic, albeit in a purely chicken infection and transmission model.

## 2. Materials and Methods

### 2.1. Virus Origin and Propagation

A/chicken/England/030786/2020 H5N8 HPAIV (herewith referred to as H5N8-2020) was successfully isolated via standard methods [16] using 9–10-day-old specific-pathogen-free (SPF) embryonated chicken eggs (ECEs) and represented the first H5N8 clade 2.3.4.4b virus incursion during 2020–2021 in the UK, following disease suspicion and H5N8 confirmation at an indoor broiler breeder premises on 1 and 2 November 2020, respectively. H5N8-2020 was further propagated through a second passage in ECEs (P2) and was subjected to whole-genome sequencing, see accession numbers EPI2462496-EPI2462503 (https://www.gisaid.org/, accessed on 11 September 2023). A/mute swan/England/SA14-234255/2020 H5N1 HPAIV (herewith referred to as H5N1-2020) was isolated in SPF ECEs and represented the first detection of the less frequent H5N1 subtype during the UK clade 2.3.4.4b epizootic in 2020–2021, following the collection of a mute swan carcass on 3 December 2020. H5N1-2020 was propagated through a second passage in ECEs and underwent whole-genome sequencing (EPI2463555-EPI2463562). H5N1-2020 sequence comparisons showed no genetic changes at the consensus level from sequences directly obtained from the clinical oropharyngeal swab derived from the mute swan and the P2 virus amplified in ECEs. Both H5N8-2020 and H5N1-2020 allantoic fluids were sterile-filtered (0.2 µm, Sartorius; Göttingen, Germany) and titrated in ECEs to determine the 50% egg infectious dose (EID_50_) [17]. The P2 virus stocks were diluted in sterile 0.1M pH 7.2 phosphate-buffered saline solution (PBS) to provide the inocula at the desired doses for the *in vivo* infections.

### 2.2. Ethics and Safety Statement

All *in vivo* experiments and procedures were approved by the Animal and Plant Health Agency (APHA Weybridge) Animal Welfare and Ethical Review Body to ensure compliance with European and UK legislation under the Animal (Scientific Procedures) Act 1986 (ASPA) [18], adhering to the UK Home Office project license PP7633638. Welfare inspections were carried out up to three times daily to identify humane endpoints to minimise mortality [13], whereby chickens that developed severe clinical disease were euthanised immediately via cervical dislocation. Apparently healthy chickens that survived to the end of the study were similarly culled. All procedures involving potentially H5Nx infectious materials and infected birds were carried out in the licensed BSL3 laboratory and animal housing facilities at APHA Weybridge [15,19].

### 2.3. Experimental Design: Minimum Infectious Dose (MID_50_), Intra-Species Transmission, and Pathogenesis of H5N8-2020 and H5N1-2020 in Chickens

A total of 96 high-health status layer chickens (Hy-Line Brown, Hy-Line UK Ltd., Warwickshire, UK) were used, which were swabbed and bled prior to the experiments. The M-gene reverse transcription real-time (RRT)-PCR and ELISA testing (below) excluded ongoing or previous AIV exposure.

*MID_50_ determination and onward transmission attempts*: The experimental design was the same for both the H5N8-2018 and H5N1-2020 viruses: To determine the MID_50_, eighteen chickens (four weeks old) were randomly selected per virus, divided into three groups of six, and each group was housed in individual Perspex pens ((121 cm × 121 cm = 1.46 m^2^ floor area) × 60 cm height). Three different doses were administered, namely 2, 4, and 6 log_10_ EID_50_ (referred to as the low, medium, and high doses, respectively), with each dose administered per group of six chickens, and a total volume of 100 µL H5N8-2020 or H5N1-2020 was inoculated per chicken via the intraocular/intranasal route using dropwise instillation: Approximately 80–90 µL were applied via a single nare, with the remainder applied to the eye on the same side of the chicken. For each dose group and for both viruses, transmission from directly infected (referred to as donor “D0”) chickens to in-contact (recipient “R1”) chickens was investigated through the introduction of six naive R1 chickens at 1-day post-inoculation (dpi). Following the mortality of all D0 chickens in both high-dose groups, when the evidence of some transmission to the R1 chickens was also apparent, two additional groups of six naive chickens were introduced (recipient “R2” chickens) and cohoused with the R1 chickens at 3 dpi and 4 dpi, for further attempted transmission of H5N1-2020 and H5N8-2020, respectively. In each pen, birds were housed on straw bedding with access to clean feed and drinking water, with the water replaced daily. These experiments concluded at between 11 and 16 dpi, as indicated in the Results section, when all surviving chickens which had remained apparently healthy throughout the study were culled. The dpi time points were measured relative to the time of the inoculation of the D0 chickens and may also be referenced throughout the duration of a given experiment. Where relevant, time points specifically concerning the R1 and R2 contact chickens may be denoted as days post-contact (dpc), which followed their introduction to cohousing.

*Pathogenesis investigation*: A separate room similarly housed two groups of six chickens that were inoculated with a high dose of H5N8-2020 or H5N1-2020 viruses. Two chickens from each virus group were planned to be culled at 1, 2, and 3 dpi for pathogenesis investigations, as described below.

### 2.4. Collection of Clinical and Environmental Samples followed by RNA Extraction

Buccal and cloacal swabs (Dryswab^TM^ ENT, rayon-bud; MWE Mediwire, Corsham, UK) were collected daily, with each swab stored in 1 mL of Leibovitz L-15 medium (LM) [15]. Tissues from eighteen organs were collected at post-mortem from chickens infected with high-dose H5N8-2020 and H5N1-2020 and culled at 1, 2, and 3 dpi for pathogenesis investigations; see the Results section for tissue details. Tissues were similarly obtained from opportunistic deaths, which occurred during the MID50 determination/intra-species transmission attempts. The post-mortem examination also included the recording of any gross pathological changes. The 18 tissues from sampled chickens were stored as 10% (*v*/*v*) in LM (frozen at −70 °C until required for total RNA extraction), and/or an additional 5 tissues were also collected, resulting in 23 tissues for storage in 10% (*v*/*v*) buffered formalin, which remained unfrozen for histology and immunohistochemical analysis. Environmental samples, namely drinking water, straw/litter, and faeces of a fresh (recently deposited) character, were collected daily from all dose groups for both viruses from 1 to 7 dpi, and thereafter daily until the end of the study (16 dpi) in the high-dose groups. The solid environmental specimens were suspended at 10% (*v*/*v*) in PBS and manually shaken for 2 min, followed by incubation at room temperature for 1 h. Centrifugation at 1000× *g* for 5 min was followed by supernatant removal. Fluids from swabs, tissues and environmental sample supernatants were processed for robotic viral RNA extraction (including direct total RNA extraction from drinking water) by means of a Universal Biorobot (Qiagen, Manchester, UK) [20].

### 2.5. AIV Reverse Transcription Real-Time PCR (RRT-PCR)

RNA extracted from clinical and environmental specimens was tested via the M-gene RRT-PCR using the primers and probe of [21], as described previously [19] by using Aria Mx Real-Time PCR instruments (Agilent Technologies, Stockport, UK). RRT-PCR Ct values ≤ 36 were considered AIV-positive, and subthreshold values in the range Ct 36.01–39.99 and Ct 40 (“No Ct”) were considered negative. A ten-fold dilution series of viral RNA (vRNA) extracted from titrated H5N8-2020 and H5N1-2020 was used to construct a standard curve using AriaMx software. The standard curve demonstrated PCR efficiency, which assured optimal assay performance for quantitative interpretation, with Ct values obtained from viral specimens converted to relative equivalent units (REUs) through correlations with the EID_50_/mL values of the extracted viral standard [22]. A chicken was considered infected if it experienced viral shedding, as measured by positive vRNA levels, from either (or both) the buccal and/or cloacal tracts.

### 2.6. Serology

Surviving chickens were heart bled during the cull of the low-, medium-, and high-dose groups at 11, 14, and 16 dpi, respectively. Sera derived from clotted blood samples were incubated at 56 °C for 30 min. Sera were tested by haemagglutination inhibition (HI) to detect H5-specific antibodies, using four haemagglutination units of the homologous H5N8-2020 or H5N1-2020 antigen [16]. Sera that registered a reciprocal HI titre of 1/16 or greater were considered H5-seropositive. All sera were also tested via the multispecies Influenza A ELISA (IDEXX, Montpellier, France), as per the manufacturer’s instructions, which was used for the detection of type-specific antibodies directed against the influenza A type-common nucleoprotein (NP).

### 2.7. Immunohistochemistry

Thin tissue sections were taken from organ samples fixed in 10% (*v*/*v*) buffered formalin and embedded in paraffin for investigation via both standard haemoxylin and eosin (H&E) staining and influenza A virus type-specific immunohistochemical (IHC) staining, which used a type-common anti-NP monoclonal antibody [22]. A semi-quantitative scoring system was used to describe the extent of histopathology and immunolabelling in the examined tissues.

### 2.8. Whole-Genome Sequencing (WGS) of Progeny Viruses

Viral RNA was extracted manually from the ECE-grown H5N8-2020 and H5N1-2020 P2 allantoic fluids without carrier RNA [13] and similarly from the tissue supernatants using the QIAamp Viral RNA Mini Kit (Qiagen), according to the manufacturer’s instructions. For WGS, first-strand cDNA synthesis was performed using SuperScript IV (Invitrogen) with random hexamers and the second-strand non-directional synthesis module (New England Biolabs, Hitchin, UK). cDNA was purified using AMPure beads (Beckman Coulter, High Wycombe, UK) and used for library preparation with the Nextera DNA Library Prep Kit (Illumina, Cambridge, UK) before sequencing using the NextSeq System (Illumina), all as per the manufacturers’ instructions. Paired-end Illumina reads from the inoculum and progeny viruses were assembled using a custom reference-guided alignment script (https://github.com/AMPByrne/WGS/blob/master/RefGuidedAlignment_Public.sh, accessed on 11 September 2023), and the gene sequences of progeny viruses from the infected tissues were assembled with either the H5N8-2020 (EPI2462496-EPI2462503) or H5N1-2020 (EPI2463555-EPI2463562) genome sequence as the reference. Sequence outputs were aligned using MAFFT [23] and visualised using MEGA7 [24].

## 3. Results

### 3.1. Determination of the MID_50_ and Onward Transmission among Chickens Infected with H5N8-2020 and H5N1-2020

*H5N8-2020*: None of the six D0 or six R1 chickens shed H5N8-2020 in the low-dose group (Figure 1a), demonstrating that the 2 log_10_ EID_50_ dose was insufficient to establish infection, with no vRNA detected in any of these buccal or cloacal swabs. The chickens in the low-dose group were culled at 11 dpi. In the medium- and high-dose groups, viral shedding occurred in 2/6 (33%) and 6/6 (100%) D0 chickens, respectively (Figure 1a). Therefore, the MID_50_ of H5N8-2020 in chickens was 10^4.3^ EID_50_. In the medium-dose group, none of the four remaining D0 chickens or six R1 chickens became infected, so they were culled at 14 dpi. However, in the high-dose group, H5N8-2020 shedding was detected in the same R1 chicken (1/6, 17%) at 5 and 6 dpi (4 and 5 dpc). Six contact R2 chickens were added to the high-dose group at 4 dpi after all D0 chickens in this group had succumbed to infection. All nine H5N8-2020-infected chickens in the medium- and high-dose groups died (Figure 2a). None of the R2 chickens became infected (Figure 1a) and were culled at 16 dpi. All sera drawn during the culls of the surviving D0, R1 and R2 chickens from all three dose groups were tested via both HI and ELISA. The absence of seroconversion confirmed the absence of H5N8-2020 infection among the survivors.

*H5N1-2020*: None of the six D0 or six R1 chickens shed H5N1-2020 in the low- and medium-dose groups (Figure 1b), which demonstrated the absence of infection, with no vRNA detected in any of these buccal or cloacal swabs. These D0 and R1 chickens were culled at 11 and 14 dpi, respectively. In the high-dose group, H5N1-2020 buccal and cloacal shedding occurred in all six (100%) D0 chickens. The MID_50_ of H5N1-2020 in chickens was therefore 10^5^ EID_50_ and was listed with the MID_50_ for H5N8-2020 (10^4.3^ EID_50_) along with infectivity values for other H5Nx clade 2.3.4.4 HPAIVs, similarly determined *in vivo* using chickens and ducks in other studies (Table 1). In the high-dose group, H5N1-2020 buccal and cloacal shedding occurred in one R1 chicken (1/6, 17%) at 5 and 6 dpi (4 and 5 dpc). Contact R2 chickens were added to the high-dose group at 3 dpi after all D0 chickens in this group had succumbed to infection. One R2 chicken (1/6, 17%) shed H5N1-2020 at 8–10 dpi (5–7 dpc) as evidence of transmission (albeit inefficient) to the R2 stage (Figure 1b). The eight H5N1-2020-infected chickens in the high-dose group all died (Figure 2b), although they attained greater mean cloacal shedding than the corresponding D0 H5N8-2020-infected chickens (compare Figure 1a,b, for the cloacal shedding after high dose inoculation). However, this observation was based on a non-statistical comparison of a small number of infected chickens which shed for a relatively short period, due to consequent HPAI mortality. None of the remaining R1 and R2 chickens in the high-dose group experienced any H5N1-2020 shedding and were culled at 16 dpi. All sera drawn during the culls of the surviving D0, R1 and R2 chickens in the three dose groups showed the absence of seroconversion, again confirming the absence of H5N1-2020 infection among the survivors.

### 3.2. Morbidity and Mortality of Chickens Infected with H5N8-2020 and H5N1-2020

Severe clinical signs were observed among all 17 chickens infected with H5N8-2020 and H5N1-2020. The virulent HPAI pathogenesis caused by both viruses resulted in all infected chickens being euthanised, in accordance with the humane endpoint.

*H5N8-2020*: In the medium-dose group, the two H5N8-2020-infected D0 chickens were euthanised at 3.5 and 4 dpi (Figure 2a) due to severe disease, which was preceded by mild clinical signs (huddling, ruffled feathers, and lethargy) for the former at 3.0 dpi, and in the course of 12 h, they deteriorated to severe neurological signs (loss of balance, tremors, and torticollis) to justify euthanasia. The latter D0 chicken in the medium-dose group suddenly developed severe paralysis at 4 dpi and was euthanised. In the high-dose group, 3/6 (50%) of the D0 chickens infected with H5N8-2020 displayed mild clinical signs (huddling, ruffled feathers, dropped wings, and lethargy) at 3 dpi in the morning, with two of these chickens rapidly deteriorating within six hours and were euthanised. The four remaining chickens in this group developed severe neurological clinical signs (loss of balance, tremors, torticollis, and paralysis) and were euthanised by 3.5 dpi. All six D0 chickens infected with high-dose H5N8-2020 were euthanised with a mean death time (MDT) of 3.42 days (Figure 2a). In the high-dose group, 1/6 (17%) of the R1 chickens began to develop mild clinical signs (including huddling, ruffled feathers, and closed eyes) at 6 dpi (5 dpc) and developed additional clinical signs upon observation throughout the day (dropped wings, the cyanosis of comb and wattles, and visual reduction in weight and loss of balance), prompting euthanasia at 6.5 dpi (5.5 dpc; Figure 2a).

*H5N1-2020*: In the high-dose group, four H5N1-2020-infected D0 chickens developed severe disease (dropped wings, swelling, huddling, ruffled feathers, lethargy, loss of balance, and tremors) at 2 dpi and were euthanised (Figure 2b). A fifth D0 chicken displayed mild clinical signs (huddling, ruffled feathers, and lethargy), which deteriorated by 2.5 dpi, with symptoms including closed eyes, dropped wings, loss of balance, and tremors. The sixth D0 chicken suddenly displayed severe clinical signs at 2.5 dpi during the evening, including swelling, diarrhoea, and discharge from the nose/beak. The MDT for these six high-dose D0 chickens infected with H5N1-2020 was 2.17 days (Figure 2b). The infected R1 chicken in the high-dose group displayed severe paralysis at 6 dpi (5 dpc) and was euthanised. The infected R2 chicken in the high-dose group displayed mild clinical signs (huddled, ruffled feathers, closed eyes, and dropped wings) at 9 dpi (6 dpc), which progressed to oedema, lethargy, loss of balance, and tremors at 10 dpi (7 dpc), and the infected chicken was then euthanised (Figure 2b).

### 3.3. Pathogenesis Investigation: Systemic Viral Distribution in Chickens after Infection with H5N8-2020 or H5N1-2020

In this experiment, two chickens infected with a high dose of each virus were pre-planned for culling at 1, 2 and 3 dpi. However, any onset of severe clinical signs following infection with either virus necessitated euthanasia, as indicated.

*H5N8-2020 infection*: For the six H5N8-20-infected chickens in the pathogenesis investigation, the four culled at 1 and 2 dpi displayed no obvious clinical signs, while at 3 dpi, one chicken displayed mild signs, and the second displayed severe signs, so it was euthanised. At 1 dpi, H5N8-2020 vRNA was detected in the nasal turbinates of D0 chickens via M-gene RRT-PCR from both chickens sampled at this time point, with very low detection in the feathers and caecal tonsils of one chicken (Figure 3a). An increased vRNA level at 2 dpi was detected in the brain, heart, kidney, and pancreas. By 3 dpi, vRNA was detected systemically in many organs. The highest relative equivalent unit (REU) titres were detected at 3 dpi in the brain (REU of 1.2 × 10^7^ EID_50_), pancreas (REU of 8.2 × 10^5^ EID_50_), and heart (REU of 7.5 × 10^5^ EID_50_) from the chicken that displayed severe signs and was euthanised (chicken #11, Figure 3a). High vRNA titres were also detected in the brain, pancreas, and heart from H5N8-2020 opportunistic deaths following infection with high (Appendix A) and medium (Appendix A) doses of H5N8-2020, which required euthanasia during the MID_50_/transmission investigation. Gross pathology at 1 dpi included mild splenomegaly and reddened pancreas, which by 2 dpi had progressed to moderate splenomegaly and moderate multifocal petechiae on the right tibial muscle. At 3 dpi, the spleen was markedly enlarged and the pancreas displayed mild multifocal mottling and pallor, suggestive of necrosis. Histopathological changes, predominantly necrotising lesions, were observed in the spleen, thymus, skin, heart, and brain by 2 dpi, and by 3 dpi, had extended to also affect the pancreas, feather follicles, and nasal turbinates (Figure 4a). H5N8-2020 antigens were infrequently detected in the turbinates and spleen at 1 dpi via IHC, but were followed by systemic viral antigen distribution by 2–3 dpi, including to the brain, heart, and lung (Figure 4b and Figure 5).

In comparison to the extensive H5N1-2020 systemic dissemination in chickens, H5N8-2020 infection resulted in an overall more limited systemic dissemination of viral antigens by 3 dpi. Viral antigens were not detected in the bursa or caecal tonsil of the two sampled H5N8-infected chickens (Figure 5). Overall, the viral IHC findings were in broad agreement with the more sensitive vRNA detection via M-gene RRT-PCR (Figure 3a).

*H5N1-2020 infection*: For the six H5N1-20-infected chickens in the pathogenesis investigation, the two culled at 1 dpi displayed no obvious clinical signs. Clinical signs were apparent at 2 dpi during cull in two chickens; another required euthanasia at 2.5 dpi, and the remaining chicken was culled at 3 dpi, displaying only mild clinical signs of ruffled feathers, dropped wings, and lethargy. Viral RNA detection following infection with H5N1-2020 at 1 dpi revealed vRNA in the nasal turbinates, heart, liver, intestine, spleen, and bursa (Figure 3b). By 2 dpi, vRNA was detected across all tissues. In addition, vRNA levels were greater in the organs from H5N1-2020-infected chickens than in those from H5N8-2020 in the pathogenesis investigation (Figure 3). Viral RNA levels in organs collected from opportunistic deaths in the MID_50_/transmission experiment also affirmed the greater systemic dissemination of H5N1-2020 infection in chickens (Appendix A). Gross pathology revealed mild-to-moderate splenomegaly at 1 dpi, which progressed to moderate splenomegaly at 2 dpi. By 3 dpi, marked splenomegaly and mild pancreatic necrosis were observed, with petechiae on the left inner tibia in one chicken.

Histopathological changes, typically necrotising lesions following H5N1-2020 infection, were seen in most tissues from 2 to 3 dpi (Figure 4a), with viral antigen detected via IHC in all tissues by 2 dpi (Figure 4b and Figure 5). The H5N1-2020 virus had systemically disseminated more rapidly in chickens and to higher levels than the H5N8-2020 virus. The IHC observations affirmed the virulence differences (including the MDT differences noted earlier) between the two clade 2.3.4.4 subtypes in chickens and were in overall accord with the findings from organ vRNA testing (Figure 3, Figure 4 and Figure 5 and Appendix A).

### 3.4. Environmental Testing

Three types of environmental samples, namely drinking water, straw/litter, and faeces, were collected daily from the high-dose pens used to determine the MID_50_ and accompanying transmission attempts, for each virus. A total of 90 environmental specimens were collected from each of the H5N8-20 and H5N1-20 housing areas from 1 to 7 dpi in the low- and medium-dose groups and daily until the end of the study at 16 dpi in the high-dose group. Following total RNA extraction from all 90 specimens, M-gene RRT-PCR revealed the absence of any environmental contamination by either the H5N8-2020 or H5N1-2020 viruses.

### 3.5. Viral Genetic Polymorphisms from Chickens Infected with H5N8-2020 and H5N1-2020

Clinical specimens, which included viral progeny from H5N8-2020 and H5N1-2020-infected tissues, were selected for WGS based on strong positive M-gene RRT-PCR results (Table 2 and Table 3). These tissue specimens largely included the brain, but also the heart and pancreas. Full sequence coverage was obtained across all eight viral genetic segments. The initial H5N8-2020 inoculum administered to the D0 chickens included mixed amino acid populations at three positions in three genes, namely PB2 (202M/V), PB1 (178E/G), and NS1 (209D/G), but viral progeny at the D0 and R1 stages demonstrated a degree of selection for the latter residue in these three genes (Table 2). No polymorphisms emerged among the H5N1-2020 progeny of the D0 and R1 chickens (Table 3). However, two polymorphisms compared with the inoculum sequence were identified in the R2 H5N1-2020-infected chickens, namely PB1 E75G and NP A234V.

## 4. Discussion

The sixth and most recent H5N1 clade 2.3.4.4 HPAIV epizootic has been ongoing in Europe since autumn 2021 and has spread to the Western Hemisphere via wild bird migration [29,30,31]. Amidst the continuing imperative to characterise the ongoing poultry threat posed by the clade 2.3.4.4b viruses, we investigated and compared the H5N1-2018 and H5N1-2020 subtypes, which featured during the immediately preceding 2020-2021 European epizootic. H5N8-2020 represented the majority subtype, while H5N1-2020 represented a less frequently encountered clade 2.3.4.4b subtype, which cocirculated with other minority subtypes during 2020–2021 in Europe [32]. H5N8-2020 and H5N1-2020 differed not only in their NA genetic segments but also in five of their internal genes (PB2, PB1, PA, NP, and NS); hence, only two of the eight viral genetic segments were closely related, namely segments 4 (HA [H5] gene) and 7 (M-gene) [9,33]. The two UK H5N8 and H5N1 isolates from late 2020 therefore represented quite distinct genotypes, and both were assessed and compared for their infectivity, transmission, and pathogenesis in chickens. Subsequent events have shown that H5N1-2020 was the direct ancestor of the H5N1 HPAIV, which had emerged in the UK and Europe as the dominant clade 2.3.4.4b virus by autumn 2021 [33].

Additional bioinformatic analyses of the UK poultry outbreaks during the 2020–2021 epizootic supported the likelihood that these were all due to multiple independent primary incursions from infected wild birds [33]. Therefore, there was no evidence for any farm-to-farm spread, which could have led to the emergence of chicken-adapted lineages, whether during 2020–2021 or during the subsequent (continuing) H5N1 clade 2.3.4.4b epizootic since autumn 2021. The same range of amino acid polymorphisms was observed among the H5Nx viral isolates from all avian species, regardless of whether they originated from poultry outbreaks (both galliforme and anseriforme sectors) or wild bird cases.

In view of the significant number of UK commercial layer premises that have been affected by H5N1 clade 2.3.4.4b HPAIVs since autumn 2021 [34,35], these *in vivo* investigations were carried out using a commercial layer breed. However, due to welfare considerations concerning the permitted housing density of the chickens, which is governed by their size (age), it was not possible to use adult point-of-lay hens; hence, 4-week-old layers were used. The MID_50_ determinations for the H5N8-2020 and H5N1-2020 showed that both viruses possessed generally similar infectivity in layers compared with other European H5Nx clade 2.3.4.4 HPAIVs (Table 1). At the beginning of the extensive North American H5N2 clade 2.3.4.4 epizootic in late 2014, the initial H5N2-2014 HPAIV was detected in December 2014, along with its immediate H5N8-2014 progenitor. Both subtypes were of wild bird origin, with both believed to have been isolated relatively soon after migratory bird incursion into the USA, and both were shown to be infectious in chickens with broadly similar MID_50_ values to those of the subsequent European H5Nx clade 2.3.4.4 HPAIVs [28]. The H5N8-2020 and H5N1-2021 isolates for the clade 2.3.4.4b *in vivo* investigations described in the current manuscript were selected according to a similar criterion of being the UK index cases for both subtypes during autumn 2020.

Returning to the H5N2 HPAIV, which then spread across the US during 2015, three of the four subsequent H5N2 North American isolates were shown to have become more infectious in chickens [27] (summarised in Table 1). It should be noted that these three H5N2-2015 viruses were isolated from the numerous H5N2 clade 2.3.4.4 outbreaks in commercial galliforme premises in the US over a period of several months. The increased chicken infectivity indicated that the H5N2-2015 virus had experienced a stronger adaptation to chickens during the US 2015 epizootic, which predominantly affected commercial galliforme poultry, with little obvious evidence of significant onward dissemination via wild birds [27]. The quantification of MID_50_ values in chickens is therefore important during prolonged outbreaks that are restricted to commercial galliformes to see whether continued incursions in terrestrial poultry may eventually result in higher infectivity and adaptation to chickens. It is speculated that the lower MID_50_ value observed in chickens infected with H5N8-2020 compared to H5N1-2020 may have been partly influenced by its broiler-breeder origins, although (as noted above) no chicken-specific lineages emerged during the 2020–2021 epizootic in the UK [33]. However, MID_50_ values of H5Nx clade 2.3.4.4 HPAIVs in ducks have revealed that these viruses are clearly more infectious in this host than in chickens (Table 1), thereby underlining their essential waterfowl-adapted nature, which reflects their propensity for their continuous dissemination and maintenance in anseriformes [13,25,26,27].

The intra-species transmission of both these clade 2.3.4.4b viruses among layer chickens was investigated. The transmission frequency appeared to be inefficient, with the H5N8-20 infecting only one of the six (17%) R1 contacts in the high-dose group, whereas the initially infected D0 chickens were all (100%) infected, with no onward transmission to the R2 contact stage at all (0% transmission). The medium-dose pen housed only two H5N8-2020-infected D0 chickens, where this lower number of shedding chickens did not result in any transmission to the cohoused R1 chickens. The transmission of H5N1-2020 similarly occurred inefficiently from the D0 chickens (again, all showed H5N1-2020 shedding) to only a single (17%) contact R1 chicken, although onward spread to the subsequent R2 contact group did occur, where again only one (17%) chicken became infected with H5N1-2020. In view of only (at most) a single cohoused chicken becoming infected with H5N8-2020 or H5N1-2020, these small numbers of infected contacts suggest that it is not possible to draw any firm conclusions concerning the comparative transmissibility of either virus. However, it was speculated that the greater virus levels observed during H5N1-2020 cloacal shedding compared with H5N8-2020 cloacal shedding may be a contributory factor in the former’s transmission to the R2 chicken stage.

The numbers of chickens in each transmission group are clearly very small compared with those observed during recent H5N1 clade 2.3.4.4b HPAIV outbreaks in large commercial poultry premises, which have included spread within layer chickens in the UK [35]. James et al. (2023) also investigated the chicken transmission of the clade 2.3.4.4b descendent H5N1-2021 HPAIV by using six chickens in the respective D0 and R1 groups, with initial infections carried out at different doses, along with an additional transmission attempt where 15 D0 and 15 R1 chickens were cohoused at a greater density [25]. However, no successful transmission was observed, despite using the same chicken breed at a similar age (3 weeks old) to enable comparison with the current study. There is a clear contrast between the ready spread of H5Nx clade 2.3.4.4b HPAIVs among numerous chickens densely housed within commercial premises and, at most, inefficient transmission observed during the much smaller experimental chicken studies. This difference may be simply explained by the limited chicken numbers that can be practically housed for ethically approved *in vivo* studies. In addition, factors associated with husbandry and behavioural changes experienced by birds in a commercial setting are challenging to simulate in an experimental study. Investigations of earlier European clade 2.3.4.4 HPAIVs, namely UK- (H5N8-2014) and Danish-origin H5Nx HPAIVs (H5N8-2016 and H5N6-2018) also demonstrated inefficient onward experimental transmission among chickens [13,26,36]. Chicken-to-chicken transmission was also considered to have been unlikely in experiments featuring the North American-origin clade 2.3.4.4 H5N8-2014 and H5N2-2014 HPAIVs [28]. To our knowledge, the only clearly successful description of efficient experimental transmission of an H5Nx clade 2.3.4.4 concerned a Hungarian-origin H5N8-2017 HPAIV [37]. Interestingly, the authors noted that no humane endpoints were employed, in that HPAIV transmission would be affected by the earlier removal of strongly shedding and diseased chickens. This detail may reflect the situation on a farm where sick or dead chickens remain amongst healthy chickens, thereby facilitating opportunities for transmission.

The absence of environmental contamination by both H5N8-2020 and H5N1-2020 in the current study suggested that close contact between chickens may have been the likely route of transmission, which occurred, albeit inefficiently, under the experimental conditions. However, viral environmental contamination on the scale of recent H5N1 HPAIV commercial chicken outbreaks in the UK has been noted [38], and it occurred during an earlier H5N1 clade 2.2 turkey outbreak in the UK in 2007 where infectious faeces were detected at the premises [39]. By contrast, the highly efficient transmission of H5Nx clade 2.3.4.4 HPAIVs among ducks has been consistently reported [13,26,27], which underlines the essential waterfowl-adapted properties of these viruses, irrespective of the NA subtype. Ducks infected with H5Nx HPAIV have also been shown to differ from similarly experimentally infected chickens, in that ducks are clearly responsible for the frequent and detectable contamination of the housing environment, which is associated with the onset of efficient transmission to naive cohoused ducks [14,25].

Previous transmission studies have shown that initial spread among ducks is not accompanied by the emergence of any new polymorphisms in the clade 2.3.4.4 viral genome, suggesting that the anseriforme-adapted nature of these earlier H5N8-2014 and H5N6-2017 viruses obviates the need for any immediate evolutionary changes in their genome [13,14]. However, polymorphisms were detected among the viral progeny shed from the chickens that did acquire such infection through transmission. It has been speculated that the experimentally observed emerging clade 2.3.4.4 viral variants in chickens may represent early adaptation attempts to this host species through changes in the polymerase genes, prior to the emergence of fitter variants that may both replicate in and spread among chickens [14]. The same considerations may apply to the amino acid changes in the viral ribonucleoprotein complex (polymerase and nucleoprotein (NP) genes) observed in the current study following H5N8-2020 and H5N1-2020 infection in chickens, which was accompanied by inefficient transmission (Table 2 and Table 3). Polymerase gene changes were shown to accumulate at the viral population level over several months during 2015 when the North American H5N2 clade 2.3.4.4 HPAIV spread extensively between numerous chicken and turkey premises [27]. This observation likely reflected the increasing adaptation of this H5N2 HPAIV to galliforme hosts during this earlier North American epizootic, as noted above.

The infection of the layer chickens with both H5N8-2020 and H5N1-2020 HPAIVs invariably resulted in rapid mortality, with all high-dose-infected chickens dying by 4 dpi and 3 dpi, respectively, indicating a more rapid pathogenesis for H5N1-2020 in this host. Histopathological examination revealed necrotising lesions and the systemic dissemination of these viruses to many tissues by 2 dpi, as detected via IHC staining, with stronger and more frequent viral tropism noted for the H5N1-2020 HPAIV. The more sensitive M-gene RRT-PCR testing of organs detected vRNA of both viruses as early as 1 dpi, with greater vRNA levels detected at 2 and 3 dpi, prior to chicken death. Overall, greater vRNA levels were attained in tissues affected by H5N1-2020 dissemination, compared with the H5N8-2020 vRNA levels in organs. The greater systemic dissemination and higher viral load of H5N1-2020 compared with H5N8-2020 likely explain the more rapid chicken mortality caused by H5N1-2020. Similar extensive systemic infection has been reported in chickens that were infected by earlier European-origin H5Nx clade 2.3.4.4 HPAIVs [13,14,26,40]. These observations are typical manifestations of systemic dissemination, virulent pathogenesis, and rapid mortality in chickens caused by all HPAIV strains of both H5 and H7 subtypes [41].

In summary, this study provides further information concerning the infectivity of recent European-origin H5Nx clade 2.3.4.4b HPAIVs in layer chickens. The direct comparisons featured the H5N8-2020 and H5N1-2020 HPAIVs, both of which were isolated in the UK during autumn 2020, where the latter was subsequently shown to be the direct ancestor of the H5N1 clade 2.3.4.4b, which became dominant in the UK and Europe by autumn 2021. The H5N1 clade 2.3.4.4b has continued to evolve into additional genotypes that have been detected during the subsequent year [33], showing how the reassortment of the internal viral genetic segments serves to diversify this virulent poultry pathogen. The findings in this study revealed differences in the replicative dynamism and pathogenesis between the two contemporary H5N8-2020 and H5N1-2020 genotypes, which were isolated in autumn 2020. This study illustrates how valuable information can be obtained through *in vivo* investigations of the globally important clade 2.3.4.4 H5Nx HPAIVs, particularly as they continue to evolve further through mechanisms that include both genetic reassortments to generate new genotypes, as well as through genetic drift within the genetic segments.

## Figures and Tables

**Figure 1 viruses-15-01909-f001:**
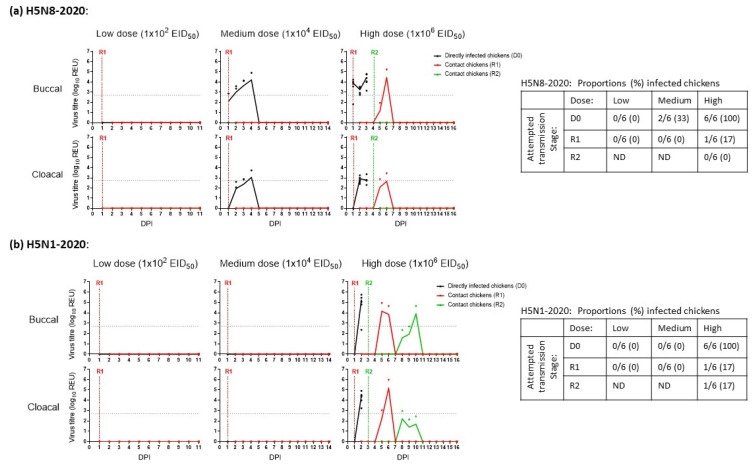
Viral shedding (buccal and cloacal) from chickens directly infected (D0) with (**a**) H5N8-2020 (A/chicken/England/030786/2020) or (**b**) H5N1-2020 (A/mute swan/England/SA14-234255/2020) at low, medium, and high doses. Continual lines and symbols indicate mean and individual shedding, respectively, in each group. Shedding from directly infected (D0), first-contact (R1), and second-contact (R2) groups are shown with black, red, and green lines/symbols, respectively. DPI indicates days post-infection, measured relative to the initial D0 inoculation day. Tables on the right summarise the proportions (%) of infected chickens in each group for H5N8-2020 (upper) and H5N1-2020 (lower), with ND (“not done”) indicating where R2 chickens were not introduced because there was no prior transmission to the R1 stage.

**Figure 2 viruses-15-01909-f002:**
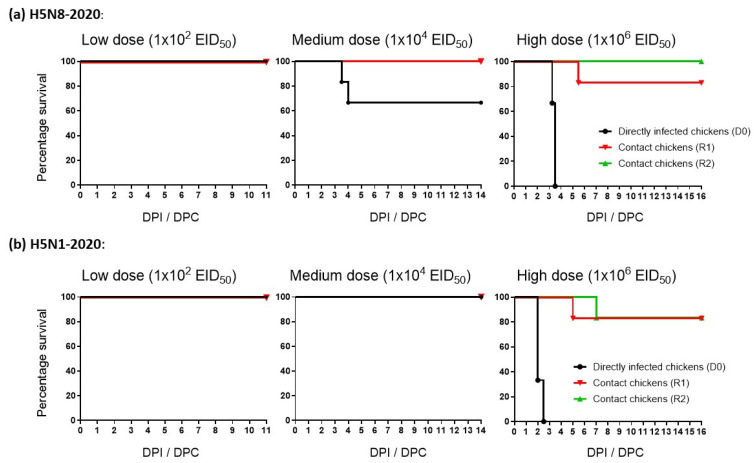
Mortality of chickens directly infected (D0; black lines) with (**a**) H5N8-2020 (A/chicken/England/030786/2020) or (**b**) H5N1-2020 (A/mute swan/England/SA14-234255/2020) at low, medium, and high doses. Mortality in the respective R1 and R2 contact groups are shown with red and green lines, respectively. DPI indicates days post-infection, while DPC indicates days post-contact for the introduced R1 and R2 chickens.

**Figure 3 viruses-15-01909-f003:**
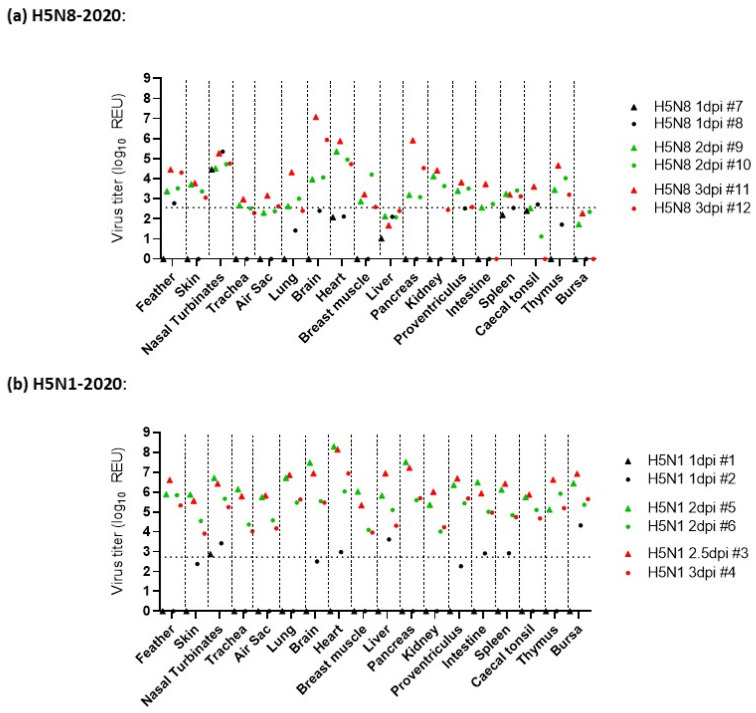
Tissue tropism (pathogenesis) comparison following H5N8-2020 and H5N1-2020 infections in chickens. Viral RNA detection is shown for 18 tissues from chickens directly infected with high-dose (**a**) H5N8-2020 and (**b**) H5N1-2020. Tissue samples were collected at 1 dpi (black symbols), 2 dpi (green symbols), and 3 dpi (red symbols, but included chicken #3, which required euthanasia at 2.5 dpi following H5N1-2020 infection).

**Figure 4 viruses-15-01909-f004:**
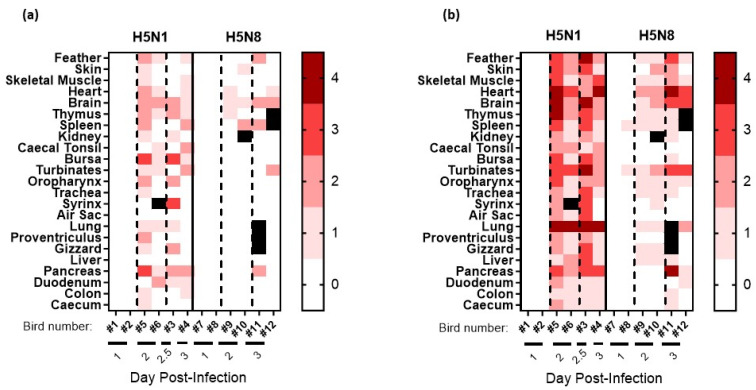
Semi-quantitative scoring of (**a**) histopathological changes visualised via H&E staining and (**b**) tissue distribution of viral-specific NP antigen determined with IHC from chickens directly infected with high-dose H5N8-2020 or H5N1-2020 during the pathogenesis time-course investigation. Tissue samples were collected at 1, 2, and 3 dpi. Viral-specific antigen staining was scored according to the following scale: 0 absent; 1 rare; 2 scattered; 3 confluent; and 4 abundant. Black squares indicate that no IHC scoring was carried out.

**Figure 5 viruses-15-01909-f005:**
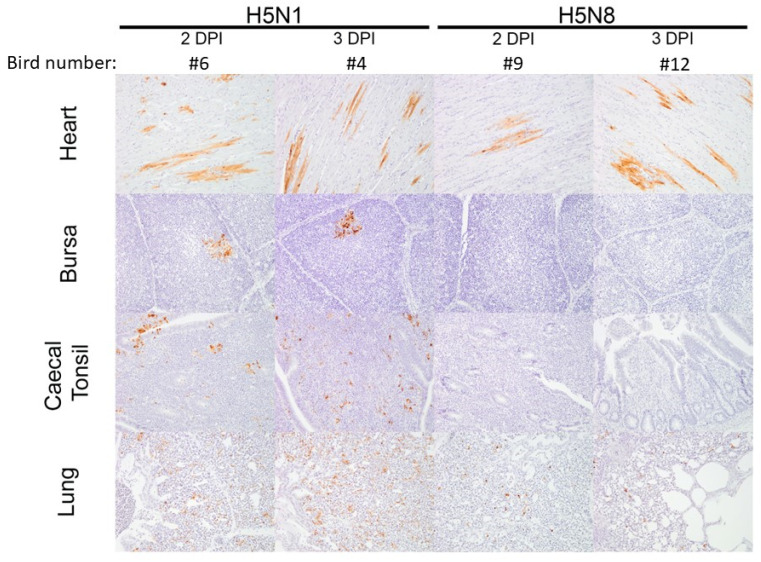
Virus-specific IHC staining observed in four tissues (heart, bursa, caecal tonsil, and lung) obtained from chickens infected with H5N8-2020 and H5N1-2020, which were sampled at 2 and 3 dpi during the pathogenesis time-course investigation.

**Table 1 viruses-15-01909-t001:** Summary of viral infectivity (50% MID_50_) values for a range of H5Nx clade 2.3.4.4 HPAIVs, as experimentally determined in chickens and ducks. The H5N8-2020 and H5N1-2020 from the current study are listed at the top, with subsequent H5Nx clade 2.3.4.4 isolates following in reverse chronological order. *Italic text* indicates three US-origin H5N2-2015 isolates from farmed galliformes, which demonstrated greater infectivity in chickens than other H5Nx clade 2.3.4.4 HPAIVs.

H5Nx Clade 2.3.4.4 HPAIV		Infectivity: (MID_50_) ^1^		Reference
Subtype-Year	Isolate Name	Layer Chickens	Ducks ^4^	
H5N8-2020	A/chicken/England/030786/2020	4.3 ^2^	ND	This manuscript
H5N1-2020	A/mute swan/England/SA14-234255/2020	5.0 ^2^	ND	
H5N1-2021	A/chicken/England/053052/2021	4.7 ^2^	<3.0	[25]
H5N6-2017	A/mute swan/England/AVP18-001986/2017	ND	3.0	[14]
H5N8-2016	A/tufted-duck/Denmark/11470/ LWPL/2016	5.0 ^3^	3.0 ^5^	[26]
H5N2-2015	A/turkey/Arkansas/7791/2015	5.1 ^3^	ND	[27]
*H5N2-2015*	*A/turkey/Minnesota/12582/2015*	*3.6 ^3^*	<2.0 ^5^	
*H5N2-2015*	*A/turkey/South Dakota/12511/2015*	*3.2 ^3^*	ND	
*H5N2-2015*	*A/chicken/Iowa/13388/2015*	*3.5 ^3^*	ND	
H5N2-2014	A/Northern pintail/Washington/ 40964/2014	5.7 ^3^	<2.0 ^5^	Chickens: [28] Ducks: [27]
H5N8-2014	A/gyrfalcon/Washington/40188–6/2014	4.4 ^3^	<2.0 ^5^	
H5N8-2014	A/duck/England/1279/2014	ND	<4.0	[13]

^1^ 50% Minimal infectious dose (MID_50_) quantified as log_10_ EID_50_/mL, calculated to one decimal point; ^2^ determined in HyLine Brown layer chickens; ^3^ determined in SPF White Leghorn layer chickens; ^4^ determined in *Anas platyrhynchos* domesticus (Pekin ducks), except where indicated by ^5^ for studies which featured *A. platyrhynchos* as mallards. ND: Not done.

**Table 2 viruses-15-01909-t002:** Summary of amino acid polymorphisms that emerged following H5N8-2020 chicken infection and transmission, with all emergent changes shown as relative to the inoculum (P2 allantoic fluids). The H5N8-2020 inoculum contained a mixture of amino acids (as indicated) within PB2, PB1, and NS1 at the stated residue positions.

Chicken Group (Transmission Stage) and Individual Identifier		Clinical Specimen	Ct Value	Genetic Polymorphisms
D0 medium dose	#71	brain	19.23	PB2: ^M^/_V_ 202V PB1: ^E^/_G_ 178G NS1: ^D^/_G_ 209D
	#72	brain	19.55	PB2: ^M^/_V_ 202M PB1: ^E^/_G_ 178E NS1: ^D^/_G_ 209D
D0 high dose	#73	brain	20.19	PB1: ^E^/_G_ 178G
	#77	brain	20.18	No changes
R1 high dose	#93	brain	21.20	PB2: ^M^/_V_ 202M PB1: ^E^/_G_ 178G NS1: ^D^/_G_ 209G

**Table 3 viruses-15-01909-t003:** Summary of amino acid polymorphisms that emerged following H5N1-2020 chicken infection and transmission, with all emergent changes shown as relative to the inoculum (P2 allantoic fluids).

Chicken Group (Transmission Stage) and Individual Identifier		Clinical Specimen	Ct Value	Genetic Polymorphisms
D0 high dose	#28	heart	17.14	No changes
R1 high dose	#45	heart	18.52	No changes
		brain	18.01	
R2 high dose	#53	pancreas	23.76	PB1: E75G NP: A234V
		brain	23.62	

## Data Availability

All data are presented within the manuscript or as Appendix A.

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
