# Peer review of "Different Outcomes of Chicken Infection with UK-Origin H5N1-2020 and H5N8-2020 High-Pathogenicity Avian Influenza Viruses (Clade 2.3.4.4b)"

_viruses, 2023, doi:10.3390/v15091909_

Round 1

Reviewer 1 Report

line 41: .. poultry farm outbreaks ....

line 54: ...poultry farm outbreaks ....

line 55: change epizootic into epidemic (and in the whole manuscript) . There has been way back discussion on the use of epidemic in strong favor of epizootic among epidemiologists, see Dohoo et al. Nature 1994; 368: 284.

line 57-58: ..wild bird cases and poultry farm outbreaks .....

-lines 73-77: to be precise and clear it is needed for the reader to know from which birds/poultry these strains are isolated from because this has impact on the interpretation of the inoculation studies.

line 78-79: ..poultry outbreaks ...

line 95-101: from this it is clear that you use a H5N8 strain that was isolated from chickens (and therefore more or less adapted to chickens) while the H5N1 strain that you use was isolated from a mute swan. 

In the manuscript you compare the results of these two strains used in your experiments (but with a very different background in original adaptation); to me it is crucial that you pay attention to the origin of the isolates (and adaptation) in the Discussion and that this may well have had considerable consequences for the results and that (to say it mildly) in a way you have been comparing apples with pears and that your conclusions with respect to the differences between the strains you see have to be much modest in the light of the use of two strains with different background adaptation in a model in which you infect chickens). To me it would have been much more logical that both strains used (H5N8-2020 and H5N1-2020) were both isolated from chickens to make a much more fair comparison.

- Line 133: you use 4 week old chickens for the experiments; these are very young birds, and not the target birds if you discuss you results in light of the tread of the HPAI H5Nx strains for the (British) poultry industry. You have to explain why you did not use chickens that were like 20 weeks old and in egg laying production.

Author Response

Dear Editor

Please refer to the uploaded PDF filer for my response to Reviewer 1.

Sincerely

Marek J Slomka

Reviewer 2 Report

This manuscript describes a well-designed and comprehensive study evaluating pathogenesis and transmission of two relevant H5Nx influenza virus strains in chickens. The authors provide a solid introduction with a robust discussion at the end. 

Minor comments:

Line 44: Suggest modifying to: "There have been four additional clade 2.3.4.4. incursions..."

Line 129: Define RRT-PCR here as it is the first use.

Line 132-133: Suggest include "18 chickens per virus (total of 36) were randomly selected..."

Line 138: Describe how the intraocular/intranasal inoculation was performed. Was 50ul delivered by each route?

Line 154: Were there any uninfected controls used for the pathogenesis study? If not, indicate why not and how comparisons to "normal" were made, particularly in terms of organ enlargement.

Lines 285-289: This sentence is a run on and very confusing. Re-write for clarity.

Lines 315 and 351 should end with a period, not a colon.

Line 504: Remove "however" from the beginning of the sentence.

In general, there are some minor English language issues, particularly with the frequent use of the word "which," which should generally be preceded by a comma.

Author Response

Dear Editor

Please refer to the uploaded PDF file for my response to Reviewer 2.

Sincerely

Marek J Slomka

Author Response

Dear Editor

Please refer to the uploaded PDF file for my response to Reviewer 3.

Sincerely

Marek J Slomka

Round 2

Reviewer 1 Report

in line 502: use likelihood instead of liklihod

I still and will disagree with your reply on the use of epizootic.  your reply: "As is well known, this suggestion has not been accepted in the subsequent years" is because your kind of researcher is keep doing that without knowing what you are talking about. But I am not going to put more energy into it, so we are going to move on.